# Nonlinear Distortion Mitigation in Multi-IF over Fiber Transmission Using Modulation-Based Adaptive Power Allocation

**Inho Ha, Hyoung-Joon Park, Soo-Min Kang and Sang-Kook Han \***

Department of Electrical and Electronics Engineering, Yonsei University, Seoul 03722, Korea; 91hainho@yonsei.ac.kr (I.H.); phjokokok@yonsei.ac.kr (H.-J.P.); roemee817@yonsei.ac.kr (S.-M.K.)
**\*** Correspondence: skhan@yonsei.ac.kr

**Abstract:** We propose a modulation-based adaptive power allocation (MBAPA) technique for nonlinear distortion mitigation in intermediate frequency over fiber (IFoF) systems. The technique allocates the spectral power of each IF band according to the required signal-to-noise power ratio (SNR) of the modulation format. To demonstrate the performance of the technique, transmission experiments were performed in 10 km and 20 km with 24-IF bands using OFDM signals. The feasibility of the proposed MBAPA technique was experimentally verified by reducing inter-modulation distortion (IMD) power and enhancing channel linearity.

**Keywords:** intermodulation distortion; multi-IFoF transmission; mobile fronthaul network; power allocation

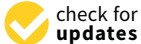



## 1. Introduction

With the advent of diverse applications and devices, the number of devices per single user and the required data capacity has been increasing significantly, leading to an exponential growth in mobile data traffic [1]. As a result, due to the limitation of data traffic per cell and the increase in the amount of data required per device, cell coverage is reduced to small cells. Many cells, including small cells, were configured to support the required data traffic. To reduce capital expenditure (CAPEX) and operating expenditures (OPEX), the radio access network (RAN) structure changed from a distributed-RAN (D-RAN) structure to a centralized RAN (C-RAN) structure in which digital units (DU) are separated from the distributed cell sites. Because communication was required between the DU and the cell site where there are only radio units (RU) available, the common public radio interface (CPRI) has been standardized and used as the digital optical transmission protocol between the DU and the RU [1,2]. However, when the signal is converted for digital transmission, the amount of transmission data is dynamically increased. For example, long term evolution (LTE, 4th generation) signal with 20 MHz bandwidth requires 2.457 Gbps speed of CPRI signal in digital transmission [2]. Likewise, the bandwidth required in 5G (5th generation) increases further, and the CPRI transmission capacity increases significantly. Thus, there is a limitation in digital transmission systems as the data required by the user increases in the current mobile fronthaul network (MFN). In order to prevent data capacity enlargement and support the required data traffic, Radio over fiber (RoF) transmission as an analog transmission has been proposed as an alternative technology [1–5]. The RoF transmission has higher spectral efficiency than the CPRI. The enhanced CPRI (eCPRI) technique that developed CPRI has been proposed. In the case of eCPRI, data enlargement is reduced due to ethernet transmission through packet-based optical transmission. However, since eCPRI technology is a packet-based ethernet transmission, it is necessary to convert the data for mobile communication. It makes the system complexity at the RU increase. In the RoF transmission, there is no process of digitizing the signal,

so the data capacity enlargement has not occurred. Accordingly, the major drawback of CPRI, data capacity enlargement, does not occur in the RoF transmission. Moreover, to use frequency resources efficiently, multiband intermediate frequency over fiber (multi-IFoF) transmission has been proposed and studied [4,6]. The RoF transmission, utilizing an intermediate carrier frequency, is the most bandwidth-efficient technique in the MFN because it preserves spectral bandwidths of the signals. For transmitting high data capacity, CPRI requires transceivers and WDM components because of data capacity enlargement, but only a single analog optical transceiver is required in the multi-IFoF transmission. Therefore, the IFoF transmission could significantly reduce the required transmission capacity [7]. Though the multi-IFoF transmission has a high-efficiency of frequency resource and has no need for an additional device to convert digital form, IFoF transmission is vulnerable to noise, especially intermodulation distortion (IMD) noise, compared to the digital transmission. Because of the difference in the receiving technique of the analog signal and the digital signal. Several techniques have been used in previous research to reduce IMD noise effects [6–14]. One technique is the equalizer using many memory taps, another technique is to pre-distort the signal to mitigate the nonlinear channel effect, and the other is employing the additional external modulators into the system [8–11]. However, as the number of devices in the cell increases and the required bandwidth of each device gets broader, the total bandwidth and data capacity increases. The signal is affected by IMD noises, including IMD2 and IMD3, which are generated by nonlinear devices and through the fiber. Because of these complex IMD noises, it will be very difficult to estimate the channel. Therefore, it is hard to reduce signal distortions by equalization and signal pre-distortion requiring accurate channel information. These techniques are not suitable to mitigate IMD noise in MFN. Existing studies have conducted just tone experiments for mitigating nonlinear components [6–9,11] or have assumed that all IF bands have the same modulation format. Furthermore, in conventional techniques, all IF bands have the same power regardless of modulation format, as shown in Figure 1. However, because each IF band supports different devices and applications, each IF band passes through different channels and has different modulation formats. After receiving the signal at the receiver, the SNR is measured and informed the channel to the transmitter.

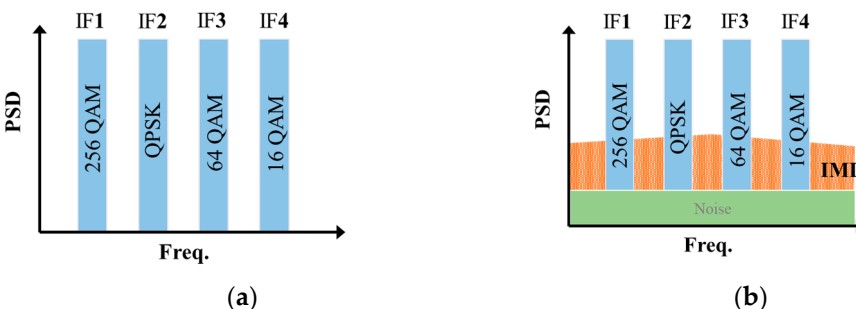

**Figure 1.** Conventional signal spectrum of (**a**) transmitted signal and (**b**) received signal.

Accordingly, the transmitter sets the modulation format, and power loading is performed with the water-pouring algorithm [15]. In the case of the CPRI, the generated orthogonal frequency-division multiplexing (OFDM) signal is converted to the time domain through the inverse Fourier transform (IFT), and the converted signal is digitized again to the baseband digital signal, then the signal is transmitted to the RU. However, since this conversion leads to explosive growth in data traffic if the OFDM signal is directly transmitted to the RU using the IFoF transmission, IMD noise is generated, and this IMD noise distorts the signal. At the RU, the distorted optical signal is downconverted to the RF frequency, and the RF signal is amplified again and transmitted to each user through the wireless channel. As shown in Figure 2a, at the RU, the distorted optical signal is downconverted to the RF frequency, and the RF signal is amplified again and transmitted to each user through the wireless channel. Since the CPRI signal is converted to the OFDM signal at

the RU, the IMD noise is not considered in the MFN using CPRI, and also, the transmitted signal power, which affects IMD noise power, was not a major consideration in the MFN. However, in the case of IFoF transmission, as shown in Figure 2b, after downconversion of the received signal, the signal is amplified without removing IMD noise at the RU, so the noise is also amplified and transmitted with the signal. Because of using an analog signal in IFoF transmission, the power of the nonlinear component affected by the signal power can be considered as the main distortion, and the generated IMD is amplified without being removed from the RU stage, which can continuously affect the signal distortion.

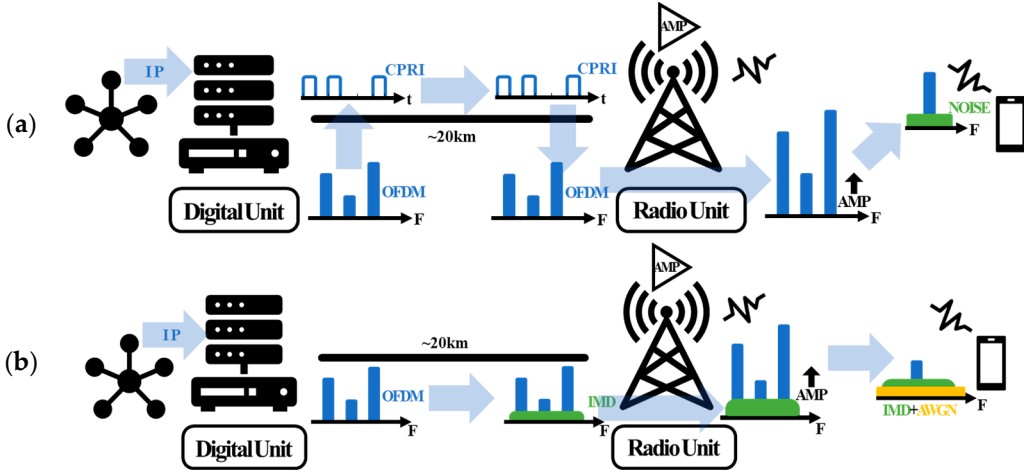

**Figure 2.** The schematic diagram of (**a**) common public radio interface (CPRI), (**b**) IF over fiber (IFoF) transmission.

In this paper, we propose a spectral power allocation technique dependent on the modulation format and verify the feasibility of system linearity enhancement of multi-IFoF-based MFN. The signal spectral power was allocated for each IF band according to modulation order, which is dependent on the required signal-to-noise power ratio (SNR), which can be converted from the error vector magnitude (EVM) requirement. Since the IMD noise power is affected by the transmitted signal power, when the total transmitted signal power is changed, the IMD noise power would be significantly changed compared to the signal power variation. We have experimentally demonstrated the reduction of IMD noise using the proposed IF power allocation. Transmission performance enhancement using the technique was verified experimentally by the EVM improvement of the received signal. This technique was briefly proposed at the Asian communication and photonics (ACP) conference 2019 and is described in detail in this paper [16].

## 2. Modulation-Based Adaptive Power Allocation

When the transmitted signal passes through the nonlinear devices or channel, the nonlinear noise is generated according to the channel transfer function, as shown in Equation (1). The transfer function has different coefficients for each different order as a function of the frequency [11,12].

$$y(t, f) = a_0 + a_1(f)x(t) + a_2(f)x(t)^2 + a_3(f)x(t)^3 + \cdots \tag{1}$$

In MFN using CPRI, the signal modulation format is determined by the signal degradation by the wireless channel between RU-Device and the fixed additive white gaussian noise (AWGN) in the entire channel, and the signal power is allocated by the water-pouring algorithm as shown in Figure 3 [17].

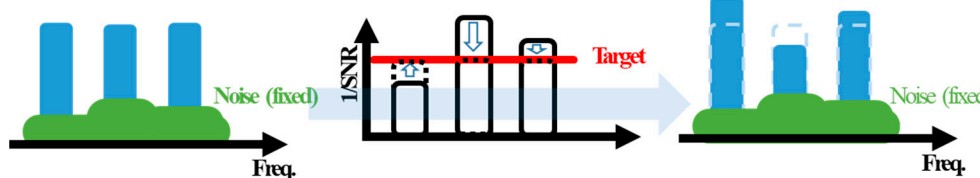

**Figure 3.** Power loading with the water-pouring algorithm.

However, in the IFoF transmission, the OFDM signals are upconverted to each IF and are transmitted to the RU, so the IMD noise and AWGN are combined and distort the signal. Since the amplifier amplifies not only the signal power also noise power, the signal is transmitted to the device with the amplified noise.

In other words, the MFN using the RoF system should transmit the signal to the RU without distortion as much as possible. In the proposed technique, the signal is generated without surplus power. It is possible to reduce the IMD noise power generated by the surplus power of the signal. As shown in Figures 4 and 5 show the spectrum of the signal passing through the nonlinear channel using the power loading and using proposed techniques.

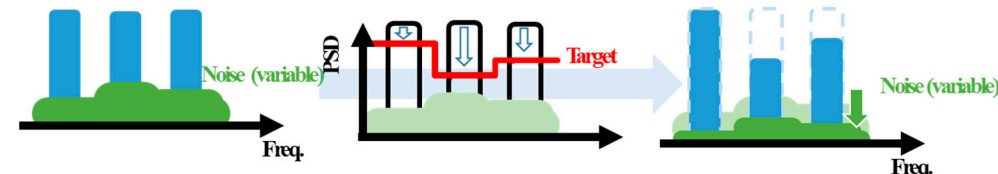

**Figure 4.** Power allocation with the proposed technique.

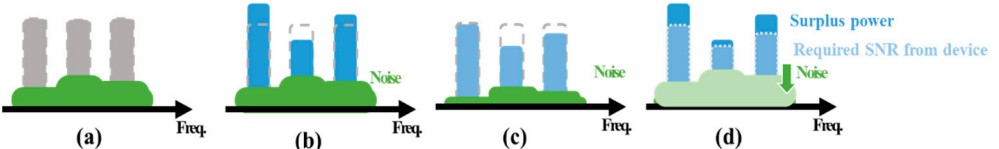

**Figure 5.** (**a**) Multiband orthogonal frequency-division multiplexing (OFDM) signal spectrum without power allocation, (**b**) signal spectrum with power loading, (**c**) signal spectrum with proposed technique, (**d**) signal spectrum comparison of power loading and proposed technique.

The IMD components are generated by device nonlinearity and fiber nonlinearity. In the case of fiber nonlinearity, the IMD2 noise is mainly the nonlinear component generated by the mixing laser chirp and chromatic dispersion when the laser is directly modulated [18,19]. The effect of dispersion is increased by increasing the transmission distance. When the directly modulated signal suffered dispersion, the IMD2 component is generated greatly [3]. Therefore, in the short distance, the IMD components caused by device nonlinearity, which are generated in RF amplifiers, a laser diode (LD), photodetector (PD), and, etc. are mainly generated as IMD3. The IMD noise power is determined by the signal power. For example, the IMD3 noise power is expressed as the product of coefficient, and the IMD3 noise power is three times of signal power [17]. That is, if the signal power decreases, the IMD3 noise power is reduced by three times the amount of the signal decrease. Therefore, a dramatic reduction in IMD noise power can be achieved by controlling the signal power [14]. In multi-IFoF transmission, there are two different IMD components. The first is the intra-band distortion generated by beating between subcarriers within a single IF band, and the other is inter-band distortion due to beating between subcarriers from other bands. In other words, in the IFoF transmission, the intra-band and inter-band IMD components are combined to generate a complex IMD component. In multi-IFoF transmission, each IF band passes through different channels and supports different devices, where each IF band signal is modulated in a different format. Because different

modulation formats have different BERs in the same SNR environment, the modulation formats require different EVM values, as shown in below Table 1 [20].

**Table 1.** Modulation format dependent error vector magnitude (EVM) and signal-to-noise power ratio (SNR) requirements in long term evolution (LTE) standard [20].

| Format | QPSK | 16QAM | 64QAM | 256QAM |
|---|---|---|---|---|
| EVM (%) | 17.5 | 12.5 | 8 | 4 |
| SNR (dB) | 15 | 18 | 22 | 28 |

The high order modulation format requires high received SNR. On the contrary, the lower-order modulation format requires a relatively low received SNR. Because of the relationship between the EVM and the SNR, required SNRs are about a minimum of 15 dB for quadrature phase-shift keying (QPSK) and 28 dB for 256 quadrature amplitude modulation (QAM) in the transmission system. The higher the modulation order, the more signal power is required. In MFN, the signal is transmitted through the fiber from DU to RU, amplified at the RU, and then sent to each device or user over the wireless channel. The dominant noise which degrades SNR occurs mainly in the nonlinear device or wireless channel. When the device receives a signal, the device periodically measures the channel. Then, the device sends a channel quality indicator (CQI) containing the measured SNR from the device. Since the modulation format of the signal is determined by the SNR from CQI [21], that is, the transmitted signal from the DU to the RU needs only an SNR that is enough to satisfy the EVM requirements until the antenna where the signal is amplified. Therefore, in the IFoF transmission, the spectral powers of the bands are allocated based on the required SNRs according to the modulation format of each band, and the allocated signal is transmitted. If the signal power per IF is allocated by the SNR required by each modulation format, the total power of the signal can be reduced, and the power of the nonlinear component is also reduced. When the SNR required by each modulation format is allocated to each IF band, the power of the signal is adjusted, and the IMD power can also be decreased, as shown in Figure 6.

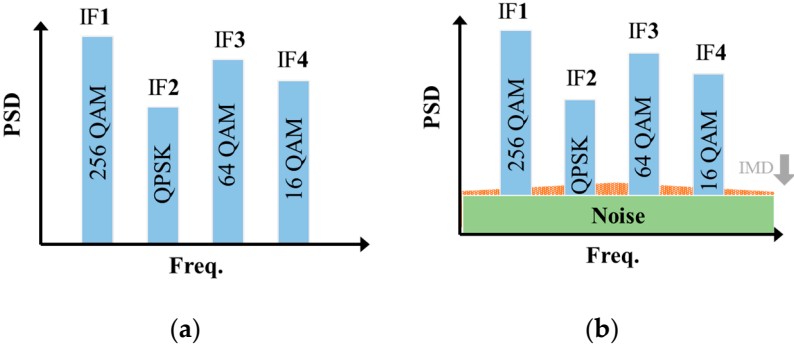

(a)                                    (b)

**Figure 6.** Signal spectrum with the proposed technique of (**a**) transmitted signal and (**b**) received signal.

The operating principle of the proposed modulation based adaptive power allocation (MBAPA) technique is as follows: When the initial power of all bands has the same power, the power of the signal band of 256QAM, the highest order, is maintained. Based on the 256QAM signal power, the excess power in each band is reduced by the required SNR difference for each order. In other words, the power of the QPSK signal is reduced by 13 dB, which is the difference from the required SNR of 256QAM. Even if the surplus power of the signal is reduced, the required EVM can be satisfied so that the signal to which the MBAPA technology to control surplus power is applied can be received.

### 3. Experiments and Results

The feasibility of the proposed MBAPA technique was verified experimentally in multi-IFoF transmission by intensity-modulated and direct detection (IM/DD). In the experiment, as shown in Figure 7, randomly generated bits were mapped to symbols depending on each modulation format of subcarriers (QPSK, 16QAM, 64QAM, and 256QAM). Bandwidth and spacing of subcarriers were set for generating OFDM signal. To have 20 MHz bandwidth per single IF, the number of subcarriers per band was 1200. The subcarrier spacing of the OFDM signal was set to be 15 kHz. The spacing between bands was set as 10 MHz, and 24 and 48 multi-IF bands were generated. The optical signal was transmitted through 10 km and 20 km optical single-mode-fiber. The received optical power at the photodetector (PD) was about −1.5 dBm, and the RF spectra of the signal, which was converted from optical to electrical signal at PD, were measured. The proposed MBAPA signal can suppress the nonlinearity of the system. It was assumed that the IMD components were generated by transfer functions of RF amplifier and LD and by power saturation at PD.

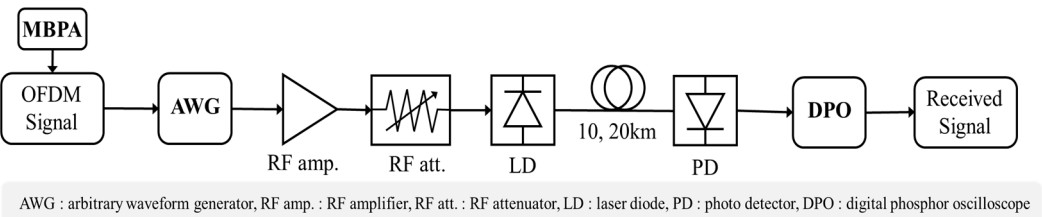

AWG : arbitrary waveform generator, RF amp. : RF amplifier, RF att. : RF attenuator, LD : laser diode, PD : photo detector, DPO : digital phosphor oscilloscope

**Figure 7.** Experimental setup.

Figure 8a,b shows the transmitted signal spectrum with the conventional technique and with the proposed technique, respectively. After 10 km transmission, two nonlinear distortions were generated and interfered with each other in a spectrum, as shown in Figure 9.

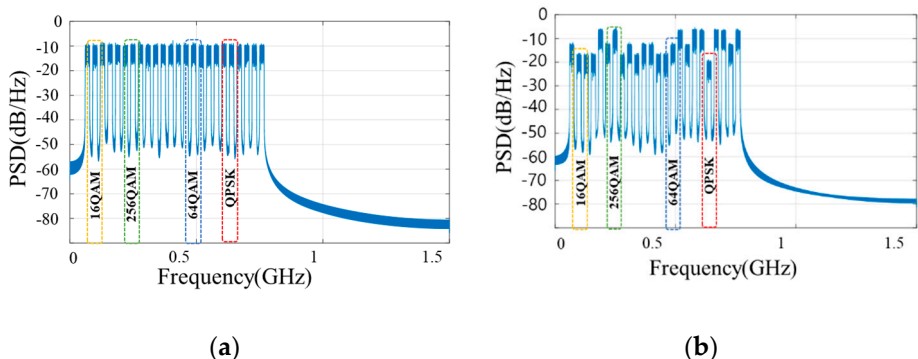

(a)  (b)

**Figure 8.** The power spectral density of transmitted signal (**a**) without the technique and (**b**) with the technique l.

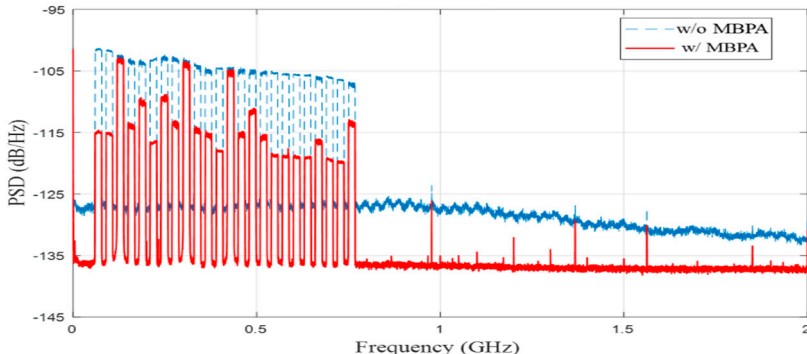

**Figure 9.** Received 24-IF bands signal spectrum after 10 km transmission.

In Figure 9, using the proposed technique, the IMD power was reduced by about 10 dB. Figure 10 shows the constellations of the first IF band, which was modulated by QPSK, and the third IF band modulated by 256QAM after a 10 km transmission with 24-IFs.

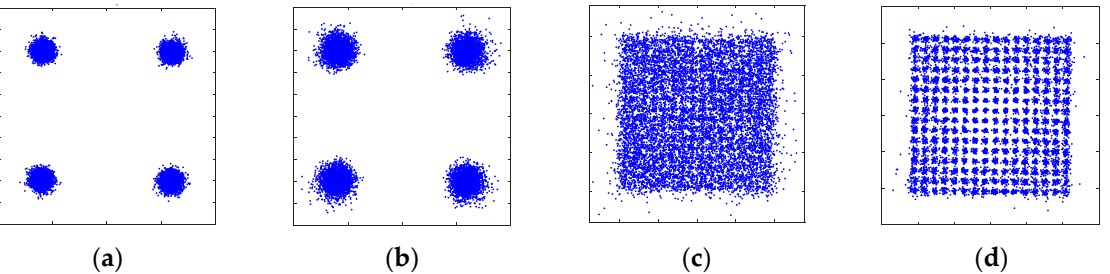

| (a) | (b) | (c) | (d) |

**Figure 10.** Constellation of the received signal after 10 km transmission with 24-IFs (**a**) quadrature phase-shift keying (QPSK) without the technique, (**b**) QPSK with the technique, (**c**) 256QAM without the technique and (**d**) 256QAM with the technique.

Figure 10a,b shows QPSK constellation diagrams, respectively, without the proposed technique and with the proposed technique. In the case of QPSK, the EVM value without the technique was 7%, and The EVM value with the technique was 11%. QPSK is a low-order modulation that requires less signal power. Therefore, the proposed technique reduces the surplus of the signal power, improves the SNR, which causes deterioration of the EVM. The EVM values of both signals satisfy the EVM requirement of the QPSK format. Figure 10c,d is the 256QAM constellation; (c) is constellation without the proposed technique, and (d) is with the proposed technique. In the case of 256QAM, unlike QPSK, the proposed technique improved EVM performance from 10.8% to 3.5%. The received signal without the technique could not satisfy the EVM requirement to receive by nonlinear distortion noise. On the other hand, with the proposed technique, the EVM requirement is satisfied, so the signal with the technique is suitable for multi-IFoF transmission. In addition, in order to verify the effect of the technique on the fiber nonlinearity, experiments were conducted by increasing the transmission distance.

The dispersion effect is more severe as increasing the transmission distance. As already written above, the dispersion mixed with the chirp effect of laser-generated the IMD2 more. As Figure 11 shows, the constellation of the 20 km transmitted signal with the same received optical power about −1.5 dBm. Similar to Figure 10, the EVM from (a) without the technique to (b) with the technique deteriorated from 6.5% to 13.4%, and both EVM values satisfied the EVM requirement. In addition, when using the proposed technique from (c) to (d), the EVM improved from 9.2% to 4%. It was proven that the IMD2 component generated by the dispersion could be effectively reduced by the proposed technique.

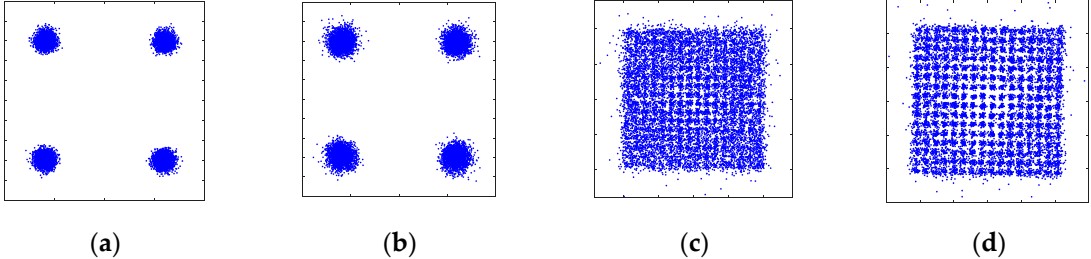

(**a**)　　　　　　　　(**b**)　　　　　　　　(**c**)　　　　　　　　(**d**)

**Figure 11.** Constellation of the received signal after 20 km transmission with 24-IFs (**a**) QPSK without the technique, (**b**) QPSK with the technique, (**c**) 256QAM without the technique and (**d**) 256QAM with the technique.

Furthermore, an experiment with increasing the total bandwidth of the transmission signal was conducted. The transmission signal was changed from a signal using 24-IF bands to a signal using 48 IF bands. As the number of IF bands used increases, the total bandwidth becomes wider, and the effects of the inter-band IMD2 and IMD3 on the signal are significantly increased. The constellation of the 48-IF signal is shown in Figure 12 after 10 km transmission. As shown in Figure 12a,b, EVM became worse from 7.1% to 10.8% using the proposed technique, but the signals satisfied the EVM requirement for QPSK. In addition, when using the proposed technique for 256QAM from (c) to (d), the EVM was improved from 10.2% to 3.4%. It was demonstrated that the proposed technique mitigates the effects of complex inter-band IMD.

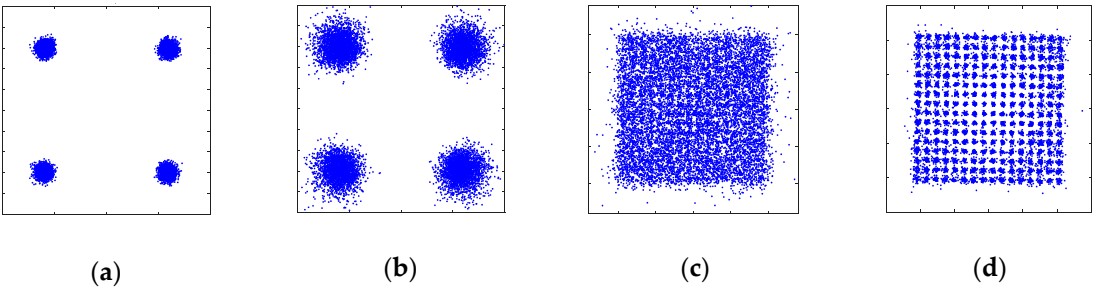

(**a**)　　　　　　　　(**b**)　　　　　　　　(**c**)　　　　　　　　(**d**)

**Figure 12.** Constellation of the received signal after 10 km transmission with 48-IFs (**a**) QPSK without the technique, (**b**) QPSK with the technique, (**c**) 256QAM without the technique and (**d**) 256QAM with the technique.

## 4. Conclusions

We proposed the modulation format based spectral power allocation technique to mitigate the IMD noise and improve channel linearity in multi-IFoF transmission. For the 24-IFoF 10 km transmission system, the proposed technique obtained 7% of EVM improvement when 256QAM was used, and also, the MBAPA technique suppressed IMD power around 10 dB. In addition, the system linearity was improved for a 48-IFoF 10 km transmission system and a 24-IFoF 10 km transmission system. The proposed technique in the reduction of IMD power effectively will be the key solution for the 5G MFN supporting multi-user.

**Author Contributions:** Conceptualization, I.H.; Data curation, I.H.; Formal analysis, I.H. and H.-J.P.; Project administration, S.-K.H.; Writing—original draft, I.H.; Writing—review & editing, S.-M.K. All authors have read and agreed to the published version of the manuscript.

**Funding:** This work was supported by the National Research Foundation of Korea (NRF) grant funded by the Korean government (MSIT; Ministry of Science and ICT) (No. NRF-2019R1A2C3007934).

**Conflicts of Interest:** The authors declare no conflict of interest.

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
