# Peer review of "Nonlinear Distortion Mitigation in Multi-IF over Fiber Transmission Using Modulation-Based Adaptive Power Allocation"

_photonics, doi:10.3390/photonics8010002_

Round 1

Reviewer 1 Report

Considering the fact that radio over optical fiber is of great interest as an alternative technology in the field of optical mobile fronthaul networks, the presented work on multiband intermediate frequency over fiber transmission and the use of modulation based adaptive power allocation to reduce nonlinear distortions in this area may be of importance and present an original approach. I recommend to publish this idea, but the paper needs several serious clarifications and improvements:

  1. The term “data explosion” is used several times in the introduction. I think this is an inappropriate formulation for this kind of scientific article. I would suggest to use “enlargement” or “increase”.
  2. In the introduction the authors mention the disadvantages of Commnon Public Radio Interface (CIPRI), but somehow they forget to mention the new standard called eCIPRI, which is more efficient than CIPRI. I will propose to make a short note to eCIPRI as well. In my opinion, this will not diminish the importance of the multiband intermediate frequency over fiber transmission.
  3. In a centralized radio access network, not only data is transmitted between the digital unit and the radio unit. Sometimes other services are also required, such as the distribution of the oscillator signal from the opto-electronic oscillator for synchronization, up-conversion and down-conversion. Perhaps the authors can also address this in the introduction. Can multi-IFoF transmission be used in a system where the oscillator is centralized in the central-station and distributed to the to multiple base-stations via a passive optical network infrastructure?
  4. The manuscript is an extension of the already published work ''Nonlinear Distortion Mitigation Technique using Modulation Format Dependent Spectral Power Allocation in Multi-IFoF system'' in Asia Communications and Photonics Conference (ACPC) 2019. The overlap and figures similarity between this manuscript and the above mentioned conference paper should be reduced. In particular, the figures should be different. The author should also explain what the difference is between this article and the paper originally published in ACPC 2019. To say that this paper describes the proposed technique in detail is not enough.
  5. All acronyms (standard and defined by the author) should be defined at the first mention. For example: 5G, PSD, QPSK, QAM, MBAPA, OFDM, AWGN.
  6. Sometimes a space is not made between the numerical value and the units, but it should always be. (e.g. page 6, line 174 “10dB”, on page 3, figure 2 “20km”, on page 5, figure 7 “20km”)
  7. It is not clear from Figure 1b what the source of the noise floor is. Why there is no noise flour in the figure 1a .
  8. Since the acronym AWGN stands for Additive White Gaussian Noise, it is not necessary to use the term “AWGN Noise”.
  9. What is the difference between MBPA and MBAPA? It is not clear why authors use two different expressions.
  10. In Figure 3 and Figure 4 there is no difference, but there should be.
  11. On page 4 in line 130 the expression “a complex IMD component” is used. The word “complex” is used several times, but it is never explained what authors mean by the word “complex”. I suspect this is not about complex numbers. It will probably be better to use the expression “high-order intermodulation products”.
  12. Since the Modulation Based Adaptive Power Allocation is the main idea of this paper. The author should explain in more detail how it works.

Author Response

We appreciate your helpful comment to make the manuscript more valuable.

We carefully read your comments and performed appropriate revisions.

Reviewer 2 Report

Review of the manuscript Photonics-1018440 “Nonlinear distortion mitigation in Multi-IF over fiber transmission using modulation based adaptive power allocation”

By Inho Ha, Hyoung-Joon Park, Soo-Min Kang, and Sang-kook Han

The authors proposed a novel technique for the mitigation of the different nonlinear distortions in the multi-intermediate frequency (IF) over fiber transmission. They demonstrated experimentally the advantages of the proposed technique for the advanced modulation formats such as QPSK, 16 QAM, 64 QAM and 256 QAM. The measured spectral characteristics and the constellation diagrams of the transmitted signals (Figures 8-12) demonstrate the high efficiency of the proposed technique. The paper is well organized and clearly written. The novel approach and experimental results obtained in this paper may be interesting for the researchers and engineers occupied in modern communication systems. I recommend the proposed paper for the publication in the present form.

Author Response

(The authors gave the same response as above.)

Reviewer 3 Report

This paper presents a nonlinear distortion mitigation scheme using modulation based power allocation technique in analog intermediate frequency-over-fiber transmission system. It should be major noise contribution that 3rd order intermodulation distortions caused by nonlinear device and composite second-order distortion in DML-based analog optical link. Using MBPA technique, the authors show that feasibility of the proposed scheme and its potential in IFoF based optical transmission system. However, the overall experimental results are quite poor to show the characteristics of the proposed technique. Only constellations and some EVM results were present.   Therefore, this paper is not suitable for publications in the current form, and the major revision should be performed before publications. Further explanations follow and would like to share it authors.

  1. What is the basis for the modulation scheme according to IF? In the case of CSO and CTB, it is possible to predict which location the distortion component will be caused when the IF arrangement is fixed. Is the modulation scheme set based on the predicted position as above? If not, what basis is the modulation scheme assigned to IF in this paper?

  2. The authors claim the proposed scheme reduces the IMD power of 10 dB in Fig. 9. However, it seems very large that the RF power injected into the laser diode when does not use MBPA. Naturally, if a large power is driven to LD or RF amplifier, the nonlinear distortion is large, but I wonder if it is correct to compare this way.

  3. Author should perform more experiments to verify the technical feasibility of proposed schemes. For example, RF driving power has a significant impact on determining the strength of nonlinear distortion.

  4. What is bias current in laser diode and RF driving power? More details about the experimental setup should be explained.

  5. The figure 10 caption number needs to be modified.

Author Response

(The authors gave the same response as above.)

Round 2

Reviewer 1 Report

My main concerns about the manuscript are answered and corrected. The paper can accept as it stands.